# Corneal Cross-Linking for Paediatric Keratoconus: A Systematic Review and Meta-Analysis

**DOI:** 10.3390/jcm10122626

**Published:** 2021-06-15

**Authors:** Hidenaga Kobashi, Osamu Hieda, Motohiro Itoi, Kazutaka Kamiya, Naoko Kato, Jun Shimazaki, Kazuo Tsubota

**Affiliations:** 1Department of Ophthalmology, School of Medicine, Keio University, Tokyo 160-8582, Japan; naokato@bc.iij4u.or.jp (N.K.); tsubota@z3.keio.jp (K.T.); 2Tsubota Laboratory Inc., Tokyo 160-0016, Japan; 3Department of Ophthalmology, Kyoto Prefectural University of Medicine, Kyoto 602-8566, Japan; ohieda@koto.kpu-m.ac.jp (O.H.); mope@koto.kpu-m.ac.jp (M.I.); 4Department of Ophthalmology, School of Medicine, University of Kitasato, Kanagawa 252-0329, Japan; kamiyak-tky@umin.ac.jp; 5Department of Ophthalmology, Ichikawa General Hospital, Tokyo Dental College, Chiba 272-8513, Japan; meishano1@gmail.com

**Keywords:** keratoconus, corneal cross-linking, pediatric

## Abstract

All corneal cross-linking techniques attenuated disease progression in patients with pediatric keratoconus for at least one year based on a meta-analysis. A standard and accelerated technique led to marked improvement in visual acuity. We determined the efficacy and safety of corneal cross-linking (CXL) in pediatric keratoconus by conducting a systematic review and meta-analysis. The PubMed and Cochrane databases were searched for relevant studies on the effects of standard, transepithelial, and/or accelerated CXL protocols in patients aged 18 years or younger. Standardized mean differences with 95% confidence intervals were calculated to compare the data collected at baseline and 12 months. The primary outcomes were maximum keratometry (Kmax) and uncorrected visual acuity (UCVA), and the secondary outcomes were the thinnest corneal thickness (TCT), best-corrected visual acuity (BCVA), and manifest refraction spherical equivalent or cylindrical refraction. Our search yielded 7913 publications, of which 26 were included in our systematic review and 21 were included in the meta-analysis. Standard CXL significantly improved the Kmax, UCVA, and BCVA, and significantly decreased the TCT. Accelerated CXL significantly improved UCVA and BCVA. In the transepithelial and accelerated-transepithelial CXL methods, each measurable parameter did not change after treatments. All CXL techniques attenuated disease progression in patients with pediatric keratoconus for at least one year. Standard and accelerated CXL led to marked improvement in visual acuity.

## 1. Introduction

Keratoconus is a progressive, frequently asymmetric, inflammatory corneal thinning disorder characterized by changes in the structure and organization of corneal collagen [1]. This progressive bilateral disease weakens the cornea, resulting in myopia, irregular astigmatism, and central corneal scarring. Keratoconus is one of the most common causes of pediatric corneal transplantation, and it accounts for approximately 15–20% of all corneal transplants in children [2]. Torres Netto EA et al. [3] reported that the prevalence of keratoconus among pediatric patients in Saudi Arabia was 4.79%, although geographical variations may exist. A young age was found to be associated with more severe forms of keratoconus and faster progression in a systematic review and meta-analysis [4]. Keratoconus progression in pediatric patients aged 18 years and younger was found to be associated with a seven-fold higher risk of requiring corneal grafting [5]. In another study, the progression of keratoconus was seen at 1 year after diagnosis in almost all of the children [6].

Corneal cross-linking (CXL) was first introduced as a promising technique to slow or stop the progression of corneal ectasia [7]. The standard method of CXL with riboflavin and ultraviolet-A (UVA, 365 nm) (3 mW/cm^2^, 30 min), now widely known as the ‘‘Dresden protocol’’, was originally developed by Wollensak et al. in Germany in 2003 [7]. The interaction of riboflavin and UVA leads to the formation of reactive oxygen species, and thus, additional covalent bonds between collagen molecules, with consequent biomechanical stiffening of the cornea [8]. After the first clinical study was published by Wollensak et al. [7], there was an increasing number of studies published on the safety and efficacy of the treatment in slowing down or halting the progression of keratoconus. There were various modifications to the standard CXL (SCXL) technique; for example, the intensity of UVA irradiation was increased, and the exposure time was shortened (accelerated CXL; ACXL) without altering the total energy delivered. Another modification is to perform CXL through an intact epithelium (transepithelial CXL; TCXL), which leads to less discomfort in the patient and fewer postoperative complications [9].

A previous systematic review and meta-analysis of CXL for the treatment of keratoconus was conducted to verify the efficacy of SCXL in stabilizing pediatric keratoconus at 1 year; however, the meta-analysis by McAnena et al. [10] included only 13 papers published prior to December 2014. To gather more evidence important for the widespread clinical use of these therapeutic techniques, we conducted this systematic review and meta-analysis of all published studies to determine the efficacy and safety of CXL in pediatric patients with progressive keratoconus.

## 2. Methods and Materials

The systematic review title and protocol were registered with PROSPERO and the Joanna Briggs Institute Database of Systematic Reviews and Implementation Reports. The meta-analysis was also performed in an academic medical setting in accordance with the Preferred Reporting Items for Systematic Reviews and Meta-analyses (PRISMA) guidelines [11].

### 2.1. Study Selection

Two reviewers searched the MEDLINE and Cochrane Central Register of Controlled Trials databases for articles published prior to 31 December 2019. Our search was performed on 15 February 2020. The keywords in our search strategy included “pediatric keratoconus”, “keratoconus”, “adolescence”, “corneal cross-linking”, “corneal collagen cross-linking”, and “collagen cross-linkage”. Two reviewers (O.H., M.I.) reviewed the titles and abstracts of the search results and retrieved the full texts of articles when the titles or abstracts appeared to meet the eligibility criteria for this review. The search strategies for MEDLINE are provided in Appendix A.

### 2.2. Inclusion and Exclusion Criteria

All observational studies examining the effects of CXL (standard, transepithelial, and/or accelerated protocols) in patients with a diagnosis of keratoconus aged 18 years or younger were included in this study. Given the paucity of available studies addressing the study question, this meta-analysis was not restricted to randomized controlled trials (RCTs); prospective and retrospective controlled clinical trials and comparative cohort studies were also included. All identified articles were carefully reviewed to select only those that reported original clinical data pre- and postoperatively. Data from the same cases included in multiple articles were omitted to exclude duplicate data. We included studies that had a minimum follow up time of one year and followed the CXL technique. When the same trial was identified by screening, we used the most recent report. Only studies including human participants published in the English language were included. We also defined the ACXL protocol to include a UVA intensity of 9 mW/cm^2^ or higher for 10 min or less. We excluded animal and ex vivo studies. Articles on CXL combined with other treatments, such as topography-guided photorefractive keratectomy and intrastromal corneal ring segments, were excluded.

### 2.3. Risk of Bias Assessment

For the observational arms of randomized controlled trials and prospective and retrospective studies, the Joanna Briggs Institute model of evidence-based healthcare bias assessment was used [12]. This model incorporates several domains, including patient sampling, randomization, inclusion, the withdrawal of patients, outcome assessment, and measurement.

### 2.4. Outcome Measures

The principle summary measure was the effectiveness of CXL in the treatment of keratoconus in patients aged 18 or younger at 1 year, and the primary outcome was the change in the maximum keratometry value (Kmax) at 1 year. To evaluate the success and failure rates, corneal flattening and steepening 1 year after CXL was defined by a change in the Kmax of more than 1.0 dioptres compared with that of the baseline value. The secondary outcomes were the change at 1 year in the thinnest corneal thickness, visual acuity, manifest refraction spherical equivalent (MRSE), cylindrical refraction, and corneal endothelial cell density. Quantitative meta-analysis was performed using the preoperative/baseline values of the primary and secondary outcomes as controls and 1-year postoperative values as the treatment group. The number of eyes that experienced adverse events postoperatively was recorded but not statistically analyzed.

### 2.5. Data Extraction and Quality Assessment

Three reviewers (H.K., O.H., M.I.) independently extracted data from the included trials and evaluated the studies based on the methods recommended in the Cochrane Handbook for Systematic Reviews of Interventions [13]. We collected data for the above outcome measures and details of the interventions, such as the setting, sample size, participants’ ages, follow up period, whether corneal epithelial removal was performed, riboflavin concentration, and UVA irradiation intensity. We requested unpublished data from the corresponding authors of individual trials via email and waited three months for a response.

### 2.6. Heterogeneity Assessment

We planned to assess heterogeneity by looking at the clinical and methodological diversity of the included studies and by examining the forest plots and I^2^ statistics, as described in the Cochrane Handbook for Systematic Reviews of Interventions [13].

### 2.7. Statistical Analysis

The baseline demographics and preoperative and 1-year postoperative mean values of the primary and secondary outcomes were combined using weighted means. The treatment effects were evaluated by the standardized mean differences (SMDs) and 95% confidence intervals (CIs) calculated for the absolute changes in the outcomes of interest. The outcomes are expressed as the mean ± standard deviation. Heterogeneity was also assessed, and an I^2^ value greater than 50% was considered significant. In the presence of significant heterogeneity, a random-effects model was used because this type of model provides a conservative estimate and is less influenced by the weighting of each study than other methods are [14]. A fixed model was used when the level of heterogeneity was less than 50%. The meta-analysis was performed using RevMan software (version 5.2, Information Management Systems Group, Cochrane Collaboration). A *p* value <0.05 was considered statistically significant using a 2-sided test.”

## 3. Results

### 3.1. Characteristics of the Included Trials

A total of 7913 articles relevant to the search terms were identified (as illustrated in Figure 1). After initial screening of titles and abstracts, there were no duplicates. We excluded 7746 studies as the secondary screening because they included case reports, review article, animal and ex vivo studies, and inadequate CXL protocols. One hundred sixty-seven articles were initially considered potentially relevant; however, 140 of these were excluded. Subsequently, based on the full texts of the remaining 27 trials, we excluded 1 trial that included mixed data for different CXL protocols. Therefore, 26 studies were included in this systematic review. As 5 studies did not report the 1-year follow-up outcomes of interest (Viswanathan et al., 2014, Shetty et al., 2014, Godefrooij et al., 2016, Chatzis et al., 2012, Padmanabhan et al., 2017), only 21 were included in the meta-analysis. The experimental and patient characteristics of the included studies are outlined in Table 1. Among these 26 trials, 17 were prospective studies [15,16,17,18,19,20,21,22,23,24,25,26,27,28,29,30,31], 7 were retrospective studies [32,33,34,35,36,37,38], and the designs of 2 studies were unclear [39,40]. A total of 1718 affected eyes were included in the systematic review. The sample sizes in these studies ranged from 10–377. These studies were performed in 11 countries (9 in Italy; 4 in India; 3 in Egypt; 2 each in Iran, the Netherlands, Switzerland, and Turkey, and 1 each in Australia, Belgium, Canada, and Chile). Of these 26 trials, 4 compared the outcomes of a few CXL protocols [27,30,31,33]. Therefore, the 31 individual trials are shown in Table 1. We classified the four techniques of CXL (19 studies included SCXL; 4 studies included ACXL, 3 studies included TCXL, and 4 studies included accelerated and transepithelial CXL (ATCXL)). One trial did not report the UVA irradiation dose or duration in the protocol [32]. The risk of bias results for the included studies are summarized in Appendix A.

### 3.2. Topographic Results

In the SCXL group (13 studies, *n* = 588 eyes), there was a significant reduction in Kmax at 1 year (SMD = 0.23; 95% CI, 0.11 to 0.34; *p* = 0.0001) (as illustrated in Figure 2A). There was a nonsignificant reduction in Kmax in the TCXL, ACXL, and ATCXL groups at 1 year (*p* = 0.98, *p* = 0.13, and *p* = 0.16, respectively) (as illustrated in Figure 2B–D). Heterogeneity was observed in the TCXL and ACXL groups (*p* = 0.05, I^2^ = 68% and *p* < 0.00001, I^2^ = 96%, respectively). The success and failure rates are shown in Table 2. In the SCXL group (six studies, *n* = 261 eyes), there was a significant decrease in the thinnest corneal thickness at 1 year (SMD = 1.31; 95% CI, 0.12 to 2.50; *p* = 0.03) (as illustrated in Figure 3A). Heterogeneity was observed in the SCXL group (*p* < 0.00001, I^2^ = 97%). The thinnest corneal thickness did not change significantly from before to after ACXL or ATCXL at the 1-year followup (*p* = 0.28 and *p* = 0.76, respectively) (as illustrated in Figure 3B,C). Heterogeneity was also observed in the ACXL group (*p* = 0.04, I^2^ = 69%). Since only one study assessed the thinnest corneal thickness in the TCXL group, the study by Buzzonetti et al. [16], we did not perform a meta-analysis of this parameter.

### 3.3. Visual Acuity and Refractive Outcomes

In the SCXL group (twelve studies, *n* = 552 eyes), there was a significant improvement in uncorrected visual acuity (UCVA) at 1 year (SMD = 0.42; 95% CI, 0.19 to 0.65; *p* = 0.0003) (as illustrated in Figure 4A). Heterogeneity was observed in the SCXL group (*p* = 0.0002, I^2^ = 69%). Similarly, there was a significant improvement in UCVA in the ACXL group at 1 year (SMD = 0.47; 95% CI, 0.11 to 0.83; *p* = 0.01) (as illustrated in Figure 4C). Heterogeneity was found in the ACXL group (*p* = 0.01, I^2^ = 72%). In the TCXL and ATCXL groups, there was a nonsignificant improvement in UCVA (*p* = 0.09 and *p* = 0.59, respectively) (as illustrated in Figure 4B,D). Heterogeneity was observed in the ATCXL group (*p* = 0.06, I^2^ = 64%).

In the SCXL group (thirteen studies, *n* = 587 eyes), there was a significant improvement in best-corrected visual acuity (BCVA) at 1 year (SMD = 0.65; 95% CI, 0.26 to 1.05; *p* = 0.001) (as illustrated in Figure 5A). Heterogeneity was observed in the SCXL group (*p* < 0.00001, I^2^ = 90%). Similarly, there was a significant improvement in BCVA in the ACXL group at 1 year (SMD = 0.47; 95% CI, 0.08 to 0.86; *p* = 0.02) (as illustrated in Figure 5C). Heterogeneity was found in the ACXL group (*p* = 0.005, I^2^ = 76%). In the TCXL and ATCXL groups, there was a nonsignificant improvement in BCVA (*p* = 0.24 and *p* = 0.64, respectively) (as illustrated in Figure 5B,D). Heterogeneity was observed in the TCXL and ATCXL groups (*p* = 0.12, I^2^ = 53% and *p* = 0.02, I^2^ = 70%, respectively).

The 7 studies included in the meta-analysis showed a significant improvement in MRSE after 1 year in the SCXL group (SMD = −0.19; 95% CI, −0.34 to −0.03; *p* = 0.02) (as illustrated in Figure 6A). However, the meta-analysis did not show differences in MRSE in any of the other groups: TCXL group (SMD = −0.22; 95% CI, −0.69 to 0.25; *p* = 0.35), ACXL group (SMD = −0.38; 95% CI, −0.81 to 0.06; *p* = 0.09) and ATCXL group (SMD = 0.17; 95% CI, −0.10 to 0.45; *p* = 0.23) (as illustrated in Figure 6B–D). Heterogeneity was observed in the ACXL group (*p* = 0.01, I^2^ = 78%).

In the SCXL group, there were no changes in cylindrical refraction over the 12 months (SMD = −0.33; 95% CI, −0.80 to 0.13; *p* = 0.16) (as illustrated in Figure 7A). Heterogeneity was observed in the SCXL group (*p* < 0.00001, I^2^ = 86%). Similarly, this meta-analysis did not TCXL group (SMD = −0.07; 95% CI, −0.54 to 0.40; *p* = 0.76), ACXL group (SMD = −0.15; 95% CI, −0.39 to 0.09; *p* = 0.22), and ATCXL group (SMD = 0.09; 95% CI, −0.14 to 0.33; *p* = 0.44) (as illustrated in Figure 7B–D).

### 3.4. Safety Outcomes

The corneal endothelial cell density remained stable in the SCXL, TCXL, and ATCXL groups at 1 year (*p* = 0.09, *p* = 0.08, and *p* = 0.81, respectively) (as illustrated in Figure 8A–C). Heterogeneity was observed in the SCXL group (*p* = 0.05, I^2^ = 63%). Since only one study, the study by Eissa et al. [31], assessed corneal endothelial cell density in the ACXL group, we did not perform a meta-analysis. The number of eyes with postoperative complications is shown in Table 2.

## 4. Discussion

We report a systematic review and meta-analysis of all CXL outcomes from 26 publications, which included 1,718 eyes of pediatric patients with progressive keratoconus. To our knowledge, this is the first comprehensive review and meta-analysis on the efficacy and safety of all CXL techniques used to treat pediatric keratoconus. The meta-analysis conducted by McAnena et al. [10] in 2017 included 13 articles on population-based prospective and retrospective studies. However, that meta-analysis did not include the accelerated and transepithelial protocol.

Kmax data were reported by all the studies that qualified for inclusion in our study. Kmax is arguably the most essential parameter when considering keratoconus progression, although it is not very reproducible. Based on our meta-analysis, Kmax significantly decreased by 0.23 D of SMD from baseline to 12 months after SCXL. McAnena et al. [10] reported a similar outcome for Kmax at the 24-month followup in an analysis of pediatric studies. Similar results were reported in reviews of studies conducted in adult patients, such as that conducted by Meiri et al. [41], which showed a 0.6 to 1 D decrease in keratometry readings at the 12–24-month followup. For ACXL, TCXL, and ATCXL, no significant differences in Kmax were observed between the baseline and 12-month postoperative visits. In a previous meta-analysis of the natural progression of keratoconus, patients younger than 17 years old and those with a baseline Kmax >55 D were at risk of at least 1.5 D progression in Kmax at 12 months. [4] In the current study, we observed evidence of no significant statistical heterogeneity in Kmax after SCXL, as indicated by an I^2^ of 16%. In terms of Kmax, considering the baseline and postoperative data, SCXL showed good potential for improving keratometry. For MRSE, this meta-analysis showed that there was a significant hyperopic shift in SCXL at 1 year. This change might be attributed to the significant flattening of Kmax.

We also found statistically significant improvements in UCVA and BCVA in the SCXL and ACXL groups, in concordance with the findings of McAnena et al. [10], who reported significant differences at 1 year. Our results also showed significant heterogeneity in those variables (I^2^ ≥ 69% at 1 year). This finding is most likely due to the within-subject variability in the measurements of UCVA and BCVA typically seen in keratoconus and secondary to irregular astigmatism. The apparent improvements in visual acuity, with such a high degree of heterogeneity, should be interpreted with caution.

In our meta-analysis, the ACXL protocol showed a significant positive effect on vision, as previous reviews have shown in adults [42,43,44]. Wen et al. [42] and Shajari et al. [43] reported comparable results between the SCXL and ACXL protocols in terms of visual acuity. However, Kobashi et al. [44] conducted a meta-analysis of RCTs and suggested that in adults, improvements in BCVA with the standard Dresden technique exceed those of the accelerated protocol. Although the accelerated protocol was not approved by the Food and Drug Administration, it is used globally and serves as a reasonable alternative to the standard protocol. Given that the accelerated technique requires shorter operative and anesthesia times, it is preferable in children who are too young to cooperate while conscious. Since the ACXL protocol is based on the Bunsen–Roscoe law of reciprocity, we assume that this principle is inapplicable for CXL in pediatric keratoconus. In adult keratoconus, the demarcation line depth after SCXL was deeper than that after ACXL according to our meta-analysis [44], which indicates that the biological effect of irradiation on a tissue differs when the total energy dose is maintained. When ACXL is used for treating eyes with keratoconus in pediatric patients, the time required to halt disease progression should be discussed.

In the current meta-analysis, Kmax and visual acuity did not change after CXL using TCXL and ATCXL. Iqbal et al. [30] also reported that ATCXL did not result in significant improvements in visual acuity or Kmax and had a success rate of only 71.6%, which was lower than that of SCXL (94.6%). However, studies on adult patients included in a meta-analysis of RCTs reported better BCVA results with the TCXL protocol than with the SCXL protocol [45]. The observed discrepancy in findings may be explained by the differences in the biomechanical properties of the cornea across age groups [46] and the paucity of relevant studies in pediatric patients treated using the transepithelial method. Although postoperative complications such as infection, sterile infiltrate, and delayed epithelial healing are more frequent with the standard epithelium-off technique, the transepithelial protocol has a smaller flattening impact on the cornea than does the standard technique. A trade-off might exist between the epithelium-off and epithelium-on techniques in terms of efficacy and safety. According to the current study, there is insufficient evidence for the use of TCXL or ATCXL in a pediatric population.

Our meta-analysis showed that the endothelial cell density did not differ between the baseline and 12-month postoperative visits with the SCXL, TCXL, and ATCXL procedures. Hence, neither a longer UV-A exposure nor a higher UV-A irradiation intensity induced pronounced endothelial cell damage. Significant heterogeneity among the four studies was found in endothelial cell density, as indicated by an I^2^ of 63% with SCXL. This heterogeneity may be induced by variability in the measurements.

This meta-analysis has at least three limitations that should be considered. First, there were no RCTs included in our meta-analysis, which led to a lower statistical power. In the present study, we included retrospective and prospective controlled studies and used the pre-CXL value of the same eye as the baseline value. Using this method, we obtained valuable data for our analysis. Another study including more prospective controlled studies is required to confirm our findings. Second, we only included data from published articles, and bias may be introduced if studies with small or different-sized effects exist but were not published. Third, long-term studies with longer follow-up periods are necessary to determine the overall efficacy and safety of CXL in pediatric patients.

In summary, all CXL techniques attenuated disease progression in patients with pediatric keratoconus for at least 1 year. Standard and accelerated CXL led to marked improvement in visual acuity. We recommend standard and accelerated CXL without documented progression in pediatric keratoconus in terms of efficacy. In contrast, the transepithelial technique is not recommended for progression attenuation in pediatric keratoconus because of insufficient efficacy.

## Figures and Tables

**Figure 1 jcm-10-02626-f001:**
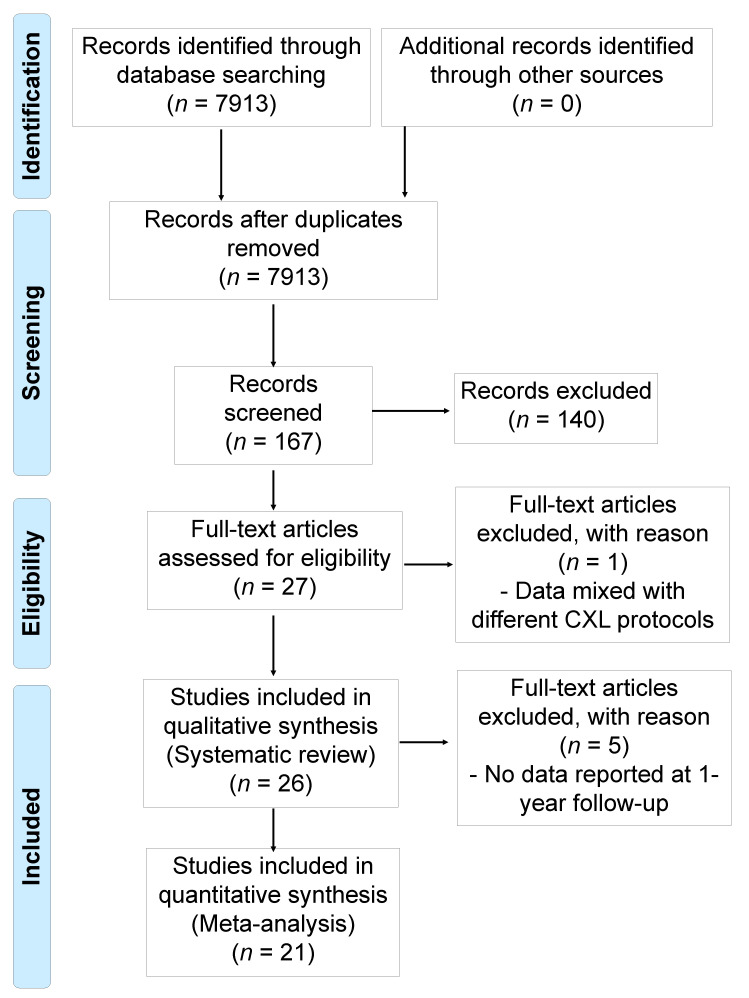
Preferred reporting items for systematic reviews and meta-analyses flow chart outlining search process to identify relevant articles from abstract identification to full paper review and inclusion of relevant publications.

**Figure 2 jcm-10-02626-f002:**
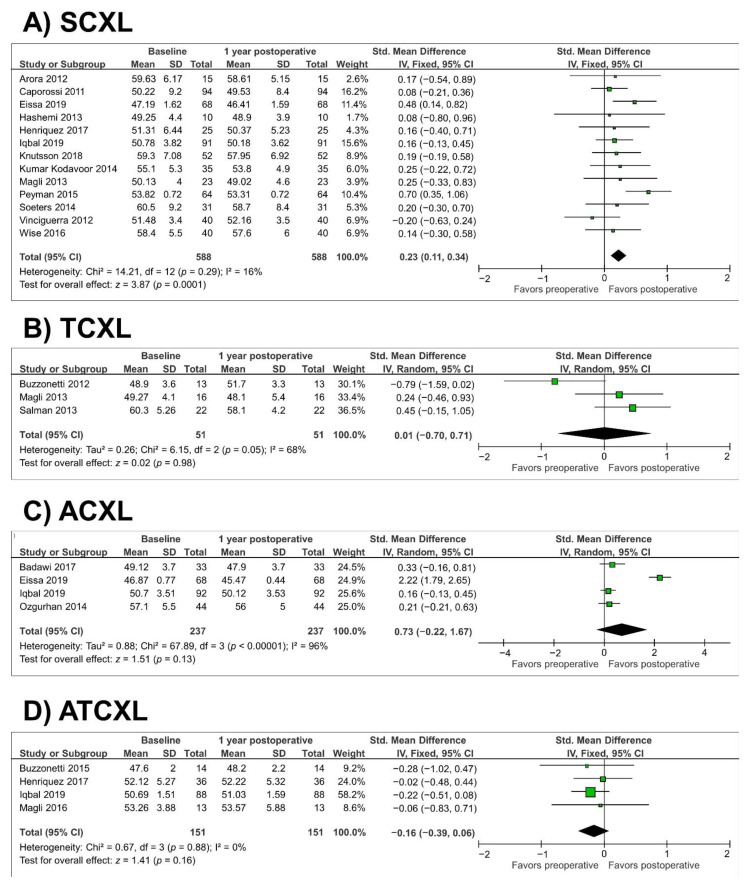
Forest plot of maximum keratometry (Kmax) 1-year standardized mean differences in dioptres in studies included in meta-analysis. (**A**), standard cross-linking (SCXL); (**B**), accelerated CXL (ACXL); (**C**), transepithelial CXL (TCXL); (**D**), accelerated and transepithelial CXL (ATCXL). IV = inverse variance, CI = confidence interval, Tau^2^ = tau-square statistic, Chi^2^ = chi-square statistic, df = degrees of freedom, I^2^ = I-square heterogeneity statistic, *z* = Z-statistic.

**Figure 3 jcm-10-02626-f003:**
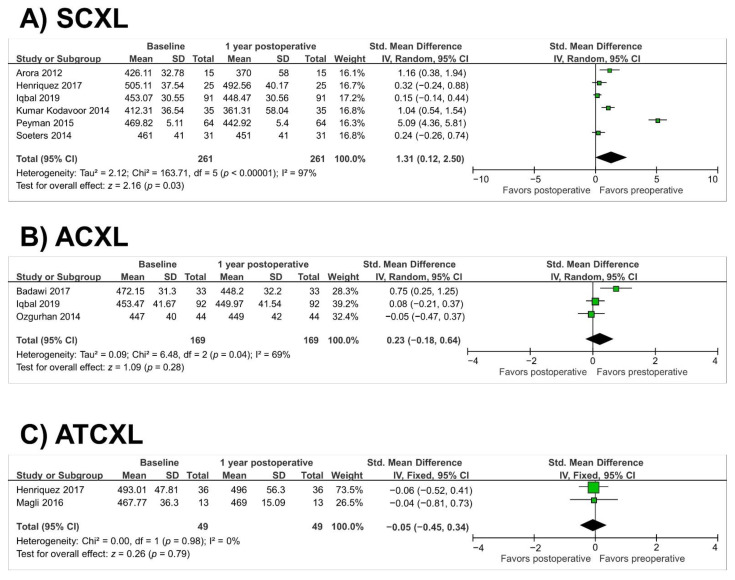
Forest plot of thinnest corneal thickness 1-year standardized mean differences in micrometres in studies included in meta-analysis. (**A**), standard cross-linking (SCXL); (**B**), accelerated CXL (ACXL), (**C**), accelerated and transepithelial CXL (ATCXL). IV = inverse variance, CI = confidence interval, Tau^2^ = tau-square statistic, Chi^2^ = chi-square statistic, df = degrees of freedom, I^2^ = I-square heterogeneity statistic, *z* = Z-statistic.

**Figure 4 jcm-10-02626-f004:**
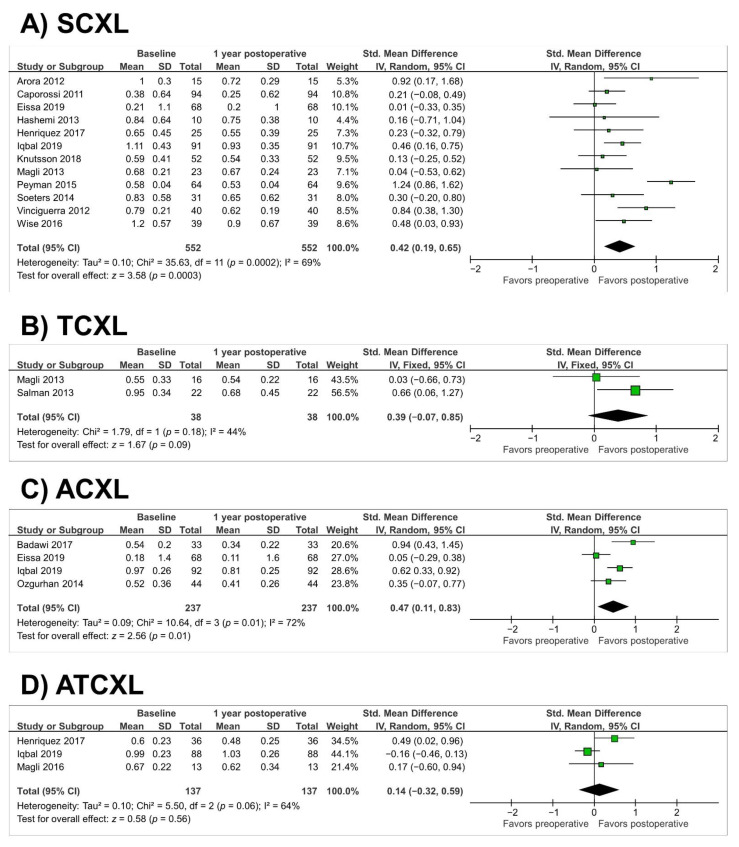
Forest plot of uncorrected visual acuity 1-year standardized mean differences in logMAR in studies included in meta-analysis. (**A**), standard cross-linking (SCXL); (**B**), accelerated CXL (ACXL); (**C**), transepithelial CXL (TCXL), (**D**), accelerated and transepithelial CXL (ATCXL). IV = inverse variance, CI = confidence interval, Tau^2^ = tau-square statistic, Chi^2^ = chi-square statistic, df = degrees of freedom, I^2^ = I-square heterogeneity statistic, *z* = Z-statistic.

**Figure 5 jcm-10-02626-f005:**
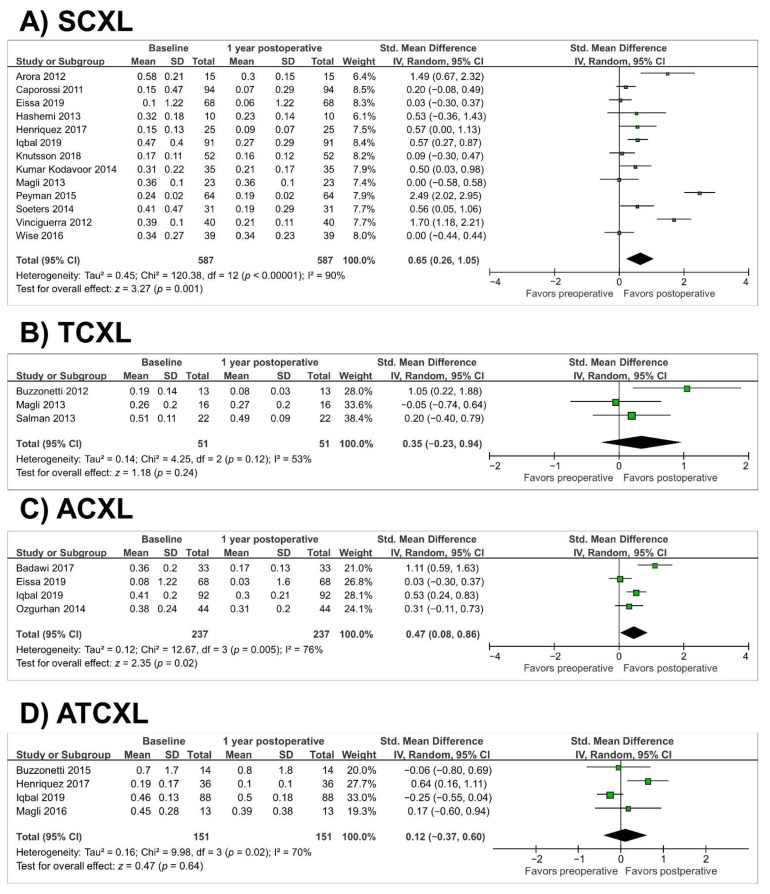
Forest plot of best-corrected visual acuity 1-year standardized mean differences in logMAR in studies included in meta-analysis. (**A**), standard cross-linking (SCXL); (**B**), accelerated CXL (ACXL); (**C**), transepithelial CXL (TCXL); (**D**), accelerated and transepithelial CXL (ATCXL). IV = inverse variance, CI = confidence interval, Tau^2^ = tau-square statistic, Chi^2^ = chi-square statistic, df = degrees of freedom, I^2^ = I-square heterogeneity statistic, *z* = Z-statistic.

**Figure 6 jcm-10-02626-f006:**
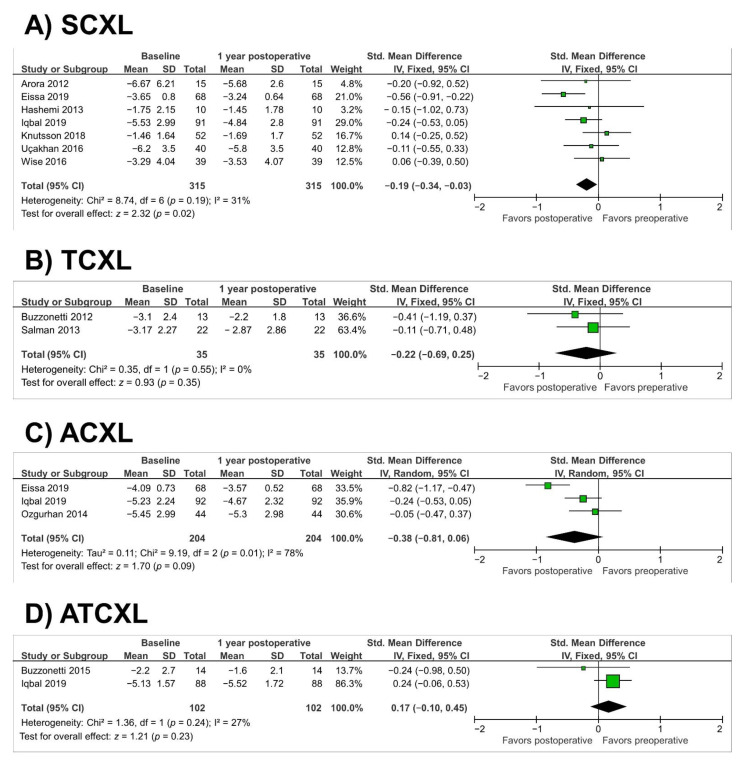
Forest plot of manifest refraction spherical equivalent 1-year standardized mean differences in dioptres in studies included in meta-analysis. (**A**), standard cross-linking (SCXL); (**B**), accelerated CXL (ACXL); (**C**), transepithelial CXL (TCXL), (**D**), accelerated and transepithelial CXL (ATCXL). IV = inverse variance, CI = confidence interval, Tau^2^ = tau-square statistic, Chi^2^ = chi-square statistic, df = degrees of freedom, I^2^ = I-square heterogeneity statistic, *z* = Z-statistic.

**Figure 7 jcm-10-02626-f007:**
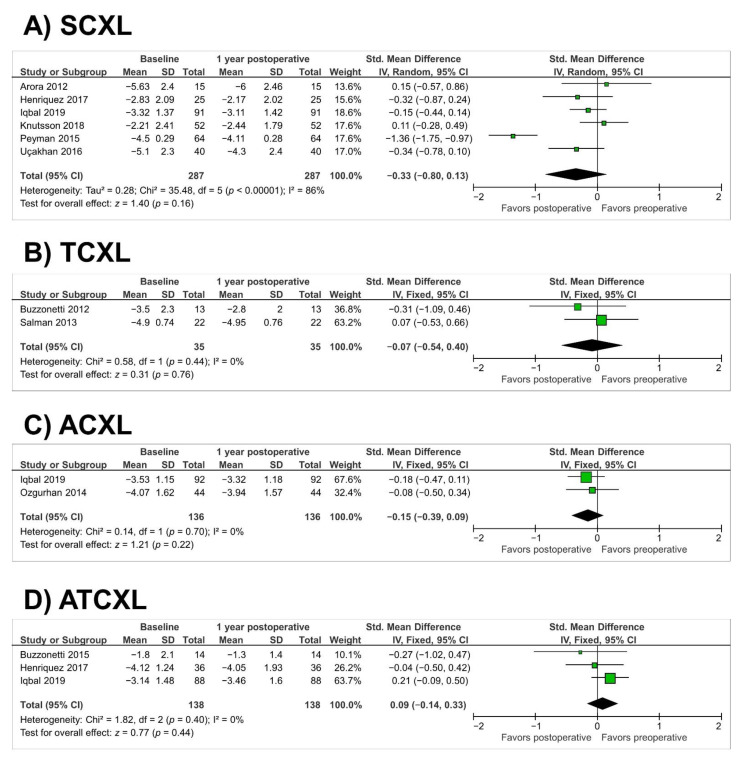
Forest plot of cylindrical refraction 1-year standardized mean differences in dioptres in studies included in meta-analysis. (**A**), standard cross-linking (SCXL); (**B**), accelerated CXL (ACXL); (**C**), transepithelial CXL (TCXL), (**D**), accelerated and transepithelial CXL (ATCXL). IV = inverse variance, CI = confidence interval, Tau^2^ = tau-square statistic, Chi^2^ = chi-square statistic, df = degrees of freedom, I^2^ = I-square heterogeneity statistic, *z* = Z-statistic.

**Figure 8 jcm-10-02626-f008:**
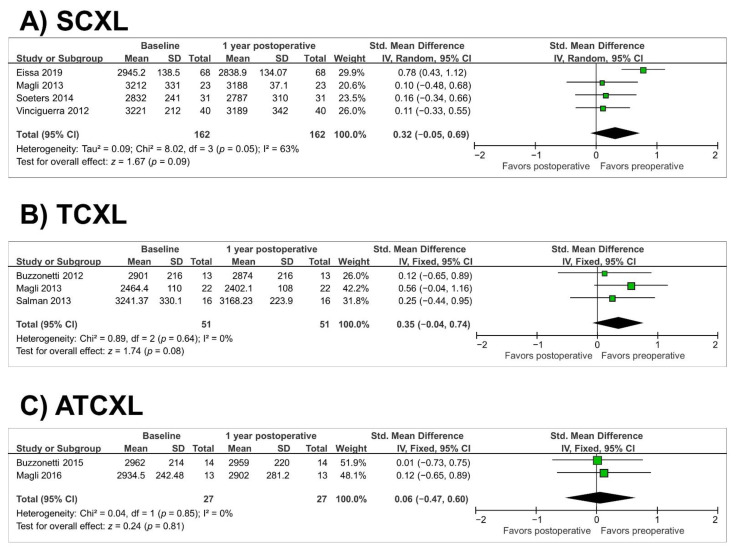
Forest plot of corneal endothelial cell density 1-year standardized mean differences in cells/mm^2^ in studies included in meta-analysis. (**A**), standard cross-linking (SCXL); (**B**), transepithelial CXL (TCXL), (**C**), accelerated and transepithelial CXL (ATCXL). IV = inverse variance, CI = confidence interval, Tau^2^ = tau-square statistic, Chi^2^ = chi-square statistic, df = degrees of freedom, I^2^ = I-square heterogeneity statistic, Z = Z-statistic.Table 2. Summary of complications of CXL after a 1-year followup.

**Table 1 jcm-10-02626-t001:** Study characteristics of all articles included in the systematic review.

First Author	Year	Country	Design	No. of Eyes	Mean Age, Yrs (SD)	Longest Follow-up (Months)	Epithelial Removal	Riboflavin Concentration (%)	UVA Irradiation (mW/cm^2^)	UVA Duration (min)	Type of CXL	Meta-Analysis
Caporossi [15]	2011	Italy	P	152	≤18	48	Off	0.1	3	30	S	☑
Buzzonetti [16]	2012	Italy	P	13	14.4 (3.7)	18	On	0.1	3	30	T	☑
Arora [17]	2012	India	P	15	≤15	12	Off	0.1	3	30	S	☑
Vinciguerra [18]	2012	Italy and Switzerland	P	40	14.2 (1.7)	24	Off	0.1	3	30	S	☑
Salman [19]	2013	Italy	P	22	15.7 (2.1)	12	On	0.1	3	30	T	☑
Chatzis [32]	2013	Switzerland	R	46	16.6	26.3	Off	0.1	NR	NR	NR	□
Hashemi [20]	2013	Iran	P	10	≤18	60	Off	0.1	3	30	S	☑
Magli [33]	2013	Naples and Italy	R	23	14.75 (2.1)	12	Off	0.1	3	30	S	☑
Magli [33]	2013	Naples and Italy	R	16	15 (2.1)	12	On	0.1	3	30	T	☑
Viswanathan [21]	2014	Australia	P	25	14.3 (2.4)	Mean: 20.1	Off	0.1	3	30	S	□
Kumar Kodavoor [34]	2014	India	R	35	13.65	12	Off	0.1	3	30	S	☑
Soeters [35]	2014	Netherlands	R	31	<18	12	Off	0.1	3	30	S	☑
Ozgurhan [36]	2014	Turkey	R	44	15.3 (2.1)	24	Off	0.1	30	4	A	☑
Shetty [22]	2014	India	P	30	12.7	24	Off	0.1	9	10	A	□
Buzzonetti [23]	2015	Italy	P	14	13.0 (2.4)	15	On (Iontophoresis)	0.1	10	9	AT	☑
Peyman [39]	2015	Iran	NR	64	15.83 (1.53)	12	Off	0.1	3	30	S	☑
Godefrooij [24]	2016	Netherlands	P	54	14.8 (1.6)	60	Off	0.1	3	30	S	□
Wise [37]	2016	Canada, Chile, and Belgium	R	39	16.3 (1.81)	12	Off	0.1	3	30	S	☑
Magli [40]	2016	Italy	NR	13	15.4 (1.7)	18	On (Iontophoresis)	0.1	10	9	AT	☑
Uçakhan [25]	2016	Turkey	P	40	15.2 (1.9)	48	Off	0.1	3	30	S	☑
Badawi [26]	2017	Egypt	P	33	12 (2.02)	12	Off	0.1	10	9	A	☑
Henriquez [27]	2017	Peru	P	25	13.2	12	Off	0.1	3	30	S	☑
Henriquez [27]	2017	Peru	P	36	14.9	12	On	0.25	18	5	AT	☑
Padmanabhan [38]	2017	India	R	377	15 (2.5)	12	Off	0.1	3	30	S	□
Knutsson [28]	2018	Italy	P	52	14.63 (2.33)	36	Off	0.1	3	30	S	☑
Mazzotta [29]	2018	Italy	P	62	14.11 (2.4)	120	Off	0.1	3	30	S	☑
Iqbal [30]	2020	Egypt	P	91	14.13 (2.18)	24	Off	0.1	3	30	S	☑
Iqbal [30]	2020	Egypt	P	92	14.4 (2.09)	24	Off	0.1	30	4	A	☑
Iqbal [30]	2020	Egypt	P	88	14.57 (2.03)	24	On	0.25	45	2.67	AT	☑
Eissa [31]	2019	Egypt	P	68	12.3 (2.4)	36	Off	0.1	3	30	S	☑
Eissa [31]	2019	Egypt	P	68	12.3 (2.4)	36	Off	0.1	18	5	A	☑

UVA = ultraviolet-A, CXL = corneal cross-linking, NR = not reported, P = prospective, R = retrospective, SD = standard deviation, S = standard, A = accelerated, T = transepithelial, AT = accelerated and transepithelial.

**Table 2 jcm-10-02626-t002:** Summary of complications of CXL after a 1-year followup.

First Author	Year	Surgical Procedures	No. of Eyes	No. of Eye with Complications						% of Eye	
				PED	Corneal Opacity	Re-Treatment	Sterile infiltrates	More than 2-Line Loss of BCVA Lines	Corneal Edema	Success Rate	Failure Rate
Caporossi [15]	2011	SCXL	152	NA	NA	NA	NA	NA	NA	NA	NA
Buzzonetti [16]	2012	TCXL	13	NA	NA	NA	NA	NA	NA	NA	NA
Arora [17]	2012	SCXL	15	NA	NA	NA	NA	NA	NA	NA	NA
Vinciguerra [18]	2012	SCXL	40	NA	NA	NA	NA	NA	NA	NA	NA
Salman [19]	2013	TCXL	22	NA	NA	NA	NA	NA	NA	NA	0
Chatzis [32]	2013	NR	46	NA	NA	NA	NA	NA	NA	65.9	11.36
Hashemi [20]	2013	SCXL	10	NA	NA	NA	NA	NA	NA	NA	NA
Magli [33]	2013	SCXL	23	NA	NA	NA	NA	NA	2	NA	NA
Magli [33]	2013	TCXL	16	NA	NA	NA	NA	NA	NA	NA	NA
Viswanathan [21]	2014	SCXL	25	NA	NA	NA	NA	NA	NA	36.0	4.0
Kumar Kodavoor [34]	2014	SCXL	35	NA	3	NA	NA	NA	NA	62.9	8.6
Soeters [35]	2014	SCXL	31	NA	NA	4	NA	1	NA	52.0	14.0
Ozgurhan [36]	2014	ACXL	44	NA	NA	NA	NA	0	NA	9.1	NA
Shetty [22]	2014	ACXL	30	0	0	NA	NA	NA	NA	10.0	NA
Buzzonetti [23]	2015	ATCXL	14	0	NA	NA	NA	NA	NA	NA	0
Peyman [39]	2015	SCXL	64	NA	NA	NA	NA	NA	NA	NA	NA
Godefrooij [24]	2016	SCXL	54	NA	NA	NA	NA	1	NA	NA	18.5
Wise [37]	2016	SCXL	33	NA	NA	NA	NA	NA	NA	NA	NA
Magli [40]	2016	ATCXL	13	NA	NA	NA	NA	NA	NA	NA	NA
Uçakhan [25]	2015	SCXL	40	NA	NA	0	2	NA	NA	NA	NA
Badawi [26]	2017	ACXL	33	NA	NA	NA	NA	0	NA	NA	NA
Henriquez [27]	2017	ATCXL	36	NA	NA	NA	1	0	NA	NA	12
Henriquez [27]	2017	SCXL	25	NA	NA	NA	0	0	NA	NA	5.6
Padmanabhan [38]	2017	SCXL	377	NA	NA	NA	0	NA	NA	52	16
Knutsson [28]	2018	SCXL	52	NA	NA	1	NA	NA	NA	NA	9.6
Mazzotta [29]	2018	SCXL	62	NA	0	NA	NA	NA	NA	NA	NA
Iqbal [30]	2020	SCXL	91	2	1	0	NA	NA	NA	NA	0
Iqbal [30]	2020	ACXL	92	0	0	2	NA	NA	NA	NA	2.2
Iqbal [30]	2020	ATCXL	88	0	0	20	NA	NA	NA	NA	9.1
Eissa [31]	2019	SCXL	68	0	0	NA	NA	NA	NA	NA	NA
Eissa [31]	2019	ACXL	68	0	0	NA	NA	NA	NA	NA	NA

CXL = corneal cross-linking, NR = not reported, SCXL = standard corneal cross-linking, ACXL = accelerated corneal cross-linking, TCXL = transepithelial corneal cross-linking, ATCXL = accelerated and transepithelial corneal cross-linking, BCVA = best-corrected visual acuity, PED = persistent epithelial defect, NA = not available.

## Data Availability

Data are available upon reasonable request.

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
