# Peer review of "Corneal Cross-Linking for Paediatric Keratoconus: A Systematic Review and Meta-Analysis"

_jcm, 2021, doi:10.3390/jcm10122626_

Round 1

Reviewer 1 Report

Dear Authors,

well prepared paper.

No issues to correct.

Very important topic - crosslinking in paediatric patients, in which progression of keratoconus is more rapid than in adults.

Moreover keratoconus is probably underdiagnosed - EA Torres Netto et al. reported prevalence of 4-5,5% in paediatic patients, although some possible geografical differences should be considered.

Best regards

Author Response

Dear Dear Editor,

Thank you for your e-mail regarding our manuscript (jcm-1204514) titled “Corneal Cross-Linking for Paediatric Keratoconus: A Systematic Review and Meta-Analysis” as well as the comments from the reviewers. We believe that the paper has been much improved, which was largely a result of the referees’ many thoughtful comments. We would like to respond below to each comment.

To Reviewer #1

Dear Authors,

well prepared paper.

No issues to correct.

Very important topic - crosslinking in paediatric patients, in which progression of keratoconus is more rapid than in adults.

Moreover keratoconus is probably underdiagnosed - EA Torres Netto et al. reported prevalence of 4-5,5% in paediatic patients, although some possible geografical differences should be considered.

Best regards

Thank you for your positive comments on our paper. We have added the comments on the prevalence of keratoconus in paediatric patients (Torres Netto EA et al. BJO. 2018).

[Page 1, lines 38-40]: One sentence has been added.

"Torres Netto EA et al.3 reported that the prevalence of keratoconus among paediatric patients in Saudi Arabia was 4.79%, although geographical variations may exist."

[Page 18, lines 361-362]: One reference has been added.

"3.         Torres Netto EA, Al-Otaibi WM, Hafezi NL, et al. Prevalence of keratoconus in paediatric patients in Riyadh, Saudi Arabia. Br J Ophthalmol. 2018;102(10):1436-1441."

We believe the manuscript has been prepared and submitted satisfactorily and hope that it will be accepted for publication in Journal of Clinical Medicine. Thank you for your attention and consideration.

Sincerely yours,

Hidenaga Kobashi, MD, PhD, Department of Ophthalmology, Keio University, School of Medicine, Tokyo, Japan. E-mail address: hidenaga_kobashi@keio.jp

Reviewer 2 Report

The text of the meta-analysis concerning CXL in keratoconus in children is written very precisely, the authors used an appropriate methodology to address the question. 

The manuscript is clear and concise, the text is very easy to follow, and it is written in an accessible manner for any reader regardless of their knowledge in the field.

Only small comments:

Abstract:

Page 1, lines 26-27: The statements of these sentences sound contratictory: „The transepithelial and accelerated-transepithelial CXL method DID NOT IMPACT ANY of the parameters.“ – this statements contradicts to the next sentence: „ALL CXL techniques ATTENUATED disease progression…..“

If a method does not change the monitored parameter („does not impact“), it cannot affect the changes (of the disease) that are measurable by this parameter…

These statements are confusing…please, reword/restyle this part and elucidate it also in the discussion part.

Results:

Figure 1:

The very clearly written text of part 3.1 of the Results is not reflected in this Figure:

Screening – Records after duplicates removed – 7913 – is the same amount as of those that were identified…does it mean, there were no duplicates? – please, elucidate this statement also in the text of the Figure 1

I miss also the information, that 7746 records were exluded in this figure…Please, explain briefly, why there were so many records excluded…

Discussion:

Page 17, line 265: splitting words at the end of a line…“ana-lysis“

Author Response

Dear Editor,

Thank you for your e-mail regarding our manuscript (jcm-1204514) titled “Corneal Cross-Linking for Paediatric Keratoconus: A Systematic Review and Meta-Analysis” as well as the comments from the reviewers. We believe that the paper has been much improved, which was largely a result of the referees’ many thoughtful comments. We would like to respond below to each comment.

To Reviewer #2

The text of the meta-analysis concerning CXL in keratoconus in children is written very precisely, the authors used an appropriate methodology to address the question.

The manuscript is clear and concise, the text is very easy to follow, and it is written in an accessible manner for any reader regardless of their knowledge in the field.

Thank you for your positive comments on our paper.

Only small comments:

Abstract:

Page 1, lines 26-27: The statements of these sentences sound contratictory: „The transepithelial and accelerated-transepithelial CXL method DID NOT IMPACT ANY of the parameters.“ – this statements contradicts to the next sentence: „ALL CXL techniques ATTENUATED disease progression…..“

If a method does not change the monitored parameter („does not impact“), it cannot affect the changes (of the disease) that are measurable by this parameter…

These statements are confusing…please, reword/restyle this part and elucidate it also in the discussion part.

We have improved the sentences to avoid the confusions as you mentioned. In the discussion part, we have elucidated the mechanism of transepithelial CXL and the difference between epi-on and epi-off.

[Page 1, lines 26-27]: One sentence has been modified.

"In the transepithelial and accelerated-transepithelial CXL methods, each measurable parameter did not change after treatments."

Results:

Figure 1:

The very clearly written text of part 3.1 of the Results is not reflected in this Figure:

Screening – Records after duplicates removed – 7913 – is the same amount as of those that were identified…does it mean, there were no duplicates? – please, elucidate this statement also in the text of the Figure 1

I miss also the information, that 7746 records were exluded in this figure…Please, explain briefly, why there were so many records excluded…

We have added the comments of Figure 1 to text. After initial screening titles and abstracts, there were no duplicates. We excluded 7,746 studies as the secondary screening because they included case reports, review article, animal and ex vivo studies, and inadequate CXL protocols.

[Page 4, lines 144-147]: One sentence has been modified and one added.

"After initial screening titles and abstracts, there were no duplicates. We excluded 7,746 studies as the secondary screening because they included case reports, review article, animal and ex vivo studies, and inadequate CXL protocols."

Discussion:

Page 17, line 265: splitting words at the end of a line…“ana-lysis“

These splitting words might be appropriately improved by JCM editorial office in the proof.

We believe the manuscript has been prepared and submitted satisfactorily and hope that it will be accepted for publication in Journal of Clinical Medicine. Thank you for your attention and consideration.

Sincerely yours,

Hidenaga Kobashi, MD, PhD, Department of Ophthalmology, Keio University, School of Medicine, Tokyo, Japan. E-mail address: hidenaga_kobashi@keio.jp

Reviewer 3 Report

Good and thorough review on pediatric keratoconus crosslinking. Well described methods and discussion section. I would recommend further building on the discussion section to highlight findings and conclusions that the authors draw. 

Stylistically, I would recommend using higher quality/resolution images and tables.

Author Response

Dear Editor,

Thank you for your e-mail regarding our manuscript (jcm-1204514) titled “Corneal Cross-Linking for Paediatric Keratoconus: A Systematic Review and Meta-Analysis” as well as the comments from the reviewers. We believe that the paper has been much improved, which was largely a result of the referees’ many thoughtful comments. We would like to respond below to each comment.

To Reviewer #3

Good and thorough review on pediatric keratoconus crosslinking. Well described methods and discussion section. I would recommend further building on the discussion section to highlight findings and conclusions that the authors draw.

Stylistically, I would recommend using higher quality/resolution images and tables.

Thank you for your positive comments on our paper.

We have improved the sentences in conclusions as you mentioned. In the discussion part, we have elucidated the mechanism of transepithelial CXL and the difference between epi-on and epi-off. The latest figures and tables were updated based on the guideline of JCM.

[Page 1, lines 26-27]: One sentence has been modified.

"In the transepithelial and accelerated-transepithelial CXL methods, each measurable parameter did not change after treatments."

We believe the manuscript has been prepared and submitted satisfactorily and hope that it will be accepted for publication in Journal of Clinical Medicine. Thank you for your attention and consideration.

Sincerely yours,

Hidenaga Kobashi, MD, PhD, Department of Ophthalmology, Keio University, School of Medicine, Tokyo, Japan. E-mail address: hidenaga_kobashi@keio.jp
